# Effect of Thermal Activation on the Structure and Electrochemical Properties of Carbon Material Obtained from Walnut Shells

**DOI:** 10.3390/ma17112514

**Published:** 2024-05-23

**Authors:** Nataliia Ivanichok, Pavlo Kolkovskyi, Oleh Ivanichok, Volodymyr Kotsyubynsky, Volodymyra Boychuk, Bogdan Rachiy, Michał Bembenek, Łukasz Warguła, Rashad Abaszade, Liubomyr Ropyak

**Affiliations:** 1Department of Material Science, Vasyl Stefanyk Precarpathian National University, 57 Shevchenko Street, 76018 Ivano-Frankivsk, Ukraine; natalia.ivanichok@gmail.com (N.I.); pavlo.kolkovskyi@pnu.edu.ua (P.K.); iomm@ukr.net (O.I.); volodymyr.kotsuybynsky@pnu.edu.ua (V.K.); volodymyra.boichuk@pnu.edu.ua (V.B.); bogdan_rachiy@ukr.net (B.R.); 2Department of Solid State Chemistry, V. I. Vernadsky Institute of General and Inorganic Chemistry, National Academy of Sciences of Ukraine, 32/34 Academician Palladin Avenue, 03142 Kyiv, Ukraine; 3Department of Manufacturing Systems, Faculty of Mechanical Engineering and Robotics, AGH University of Krakow, 30 Mickiewicza Avenue, 30-059 Kraków, Poland; 4Faculty of Mechanical Engineering, Institute of Machine Design, Poznan University of Technology, Piotrowo 3, 60-965 Poznan, Poland; 5Department of Electronics and Automations, Azerbaijan State Oil and Industry University, Azadliq Avenue 20, AZ1010 Baku, Azerbaijan; abaszada@gmail.com; 6Department of Computerized Mechanical Engineering, Ivano-Frankivsk National Technical University of Oil and Gas, 15 Karpatska Street, 76019 Ivano-Frankivsk, Ukraine; l_ropjak@ukr.net

**Keywords:** porous carbon material, thermal activation, porous structure, low-temperature porometry, fractal dimension, specific surface area

## Abstract

A simple activation method has been used to obtain porous carbon material from walnut shells. The effect of the activation duration at 400 °C in an atmosphere with limited air access on the structural, morphological, and electrochemical properties of the porous carbon material obtained from walnut shells has been studied. Moreover, the structure and morphology of the original and activated carbon samples have been characterized by SAXS, low-temperature adsorption porosimetry, SEM, and Raman spectroscopy. Therefore, the results indicate that increasing the duration of activation at a constant temperature results in a reduction in the thickness values of interplanar spacing (*d*_002_) in a range of 0.38–0.36 nm and lateral dimensions of the graphite crystallite from 3.79 to 2.52 nm. It has been demonstrated that thermal activation allows for an approximate doubling of the specific S_BET_ surface area of the original carbon material and contributes to the development of its mesoporous structure, with a relative mesopore content of approximately 75–78% and an average pore diameter of about 5 nm. The fractal dimension of the obtained carbon materials was calculated using the Frenkel–Halsey–Hill method; it shows that its values for thermally activated samples (2.52, 2.69) are significantly higher than for the original sample (2.17). Thus, the porous carbon materials obtained were used to fabricate electrodes for electrochemical capacitors. Electrochemical investigations of these cells in a 6 M KOH aqueous electrolyte were conducted by cyclic voltammetry, galvanostatic charge/discharge, and impedance spectroscopy. Consequently, it was established that the carbon material activated at 400 °C for 2 h exhibits a specific capacity of approximately 110–130 F/g at a discharge current density ranging from 4 to 100 mA/g.

## 1. Introduction

Climate change and the limited availability of fossil fuel resources compel society to transition towards sustainable renewable energy sources. As a result, we are witnessing a rise in renewable energy production from solar and wind sources, alongside the advancement of electric vehicles and hybrid electric vehicles with reduced CO_2_ emissions. However, solar energy is not available at night, and wind patterns can be unpredictable. Consequently, energy storage devices are assuming a more prominent role in our energy landscape [1,2]. While electrical energy cannot be stored directly, it can be converted into other forms of energy, such as chemical, kinetic, or potential energy, which can be stored and later converted back into electricity as needed. Among all energy storage approaches, electrochemical energy storage devices such as storage batteries and electrochemical capacitors (ECs), used in microelectronic devices, hybrid vehicles, and large industrial equipment, play a crucial role. To meet the rising energy demands of modern devices, electrochemical systems require continuous improvement through the development of new materials and a deeper understanding of the electrochemical processes occurring at the nano-level [3,4,5,6,7,8,9,10,11].

Currently, lithium-ion batteries, renowned for their high energy density, continue to lead as the most potent among energy storage devices. However, ECs are undergoing active improvement and possess a significant advantage over lithium-ion batteries, as they offer instantaneous energy supply and can undergo a large number of charge/discharge cycles [12,13,14]. Therefore, ECs play a crucial role in complementing or replacing batteries in energy conservation applications, such as serving as sources of uninterruptible power supply and voltage equalization [15].

In the production of EC electrodes, materials based on activated carbon with a high surface area are used. Even in a thin layer, such materials exhibit a total surface area significantly greater than traditional materials like aluminum, facilitating the storage of a substantial amount of charge [16,17].

Several types of composite electrodes for batteries with improved performance characteristics were obtained by plasma electrolytic oxidation of valve group metals, in particular Ti [18,19]. In works [20,21,22], methods for ensuring the high quality of such composites and ways to optimize the process of their production were studied. It should be noted that electrode porous carbon materials have several advantages.

Graphitic carbon carries out all the requirements for applications as an EC electrode material due to high conductivity, electrochemical stability, and open porosity [23]. Activated carbon, carbon derived from carbides, carbon fabrics, carbon fibers, carbon nanotubes, nanobulins, and nanohorns can all be used as electrode materials for electrochemical double-layer capacitors [24]. The most common electrode material for electrochemical capacitors is activated carbon (AC), prized for its large surface area and affordability. Natural materials such as coconut shells, wood, resins, fossil coal, or synthetic materials such as polymers are typically used as precursors for AC production. The activation process develops the porous structure within the carbon material, creating micropores (<2 nm), mesopores (2–50 nm), and macropores (>50 nm) within the carbon grains [25]. Studies of the properties of carbon materials obtained as a result of synthesis using catalysts are interesting [26,27].

Coconut shells, fruit pits, coffee grounds, stems, leaves, and other parts of plants, bamboo, or wood are most often used as bio-raw materials for obtaining porous carbon material (PCM). These materials have a high potential for use in the production of porous carbon material due to their high carbon content and the possibility of obtaining a material with the required properties of a porous structure. In particular, in the work [28], light industry wastes, namely, hemp, flax, jute, coir, and abaca fibers were investigated. It was established that the obtained carbon material can reach a surface area from 770 to 879 m^2^/g, and the yield of activated carbon was mainly less than 20 wt.% of the original biomass.

The method of obtaining PCM from bamboo cellulose fibers is proposed in [29]. Due to the unique structure (three-dimensional conductive network, hierarchical pores, high heteroatom doping), the obtained PCM as a supercapacitor electrode demonstrated excellent capacitive performance (280 F/g at 0.3 A/g), good speed, and cyclic stability. Also, one of the most common sources of biomass for obtaining PCM is wood, which contains a large amount of cellulose and lignin, which allow for obtaining PCM with high porosity. [30,31]. Due to the open-pore structure, the optimized PCM exhibits a specific capacity of 286 F/g at 1 A/g and an energy density of 5.63 W·h/kg with a power density of 9213 W/kg in 6 M KOH [30].

Porous carbon materials (PCMs) are among the most promising nanomaterials used across diverse fields of technology and industry, owing to their unique combination of physical and chemical characteristics. Modified PCM is used as electrodes in electrochemical power sources, sorbents, membranes, screens for absorbing electromagnetic radiation, and components in radio and electrical engineering. Additionally, carbon nanomaterials are currently employed in the thermal protection systems of spacecraft, airplanes, and rockets, contributing to the production of their nose and engine components [32,33,34].

Carbon nanomaterials used as electrode materials in electrochemical energy storage sources must adhere to specific criteria. These include high electrical conductivity, excellent surface wettability, controlled morphology, and pore structure characterized by specific surface area and size-distributed volume, among other factors. To obtain the necessary parameters, carbonized and graphitized carbon materials undergo thermal or chemical activation processes, which promote the development of the porous structure essential for carbon materials [35]. By adjusting the activation parameters, such as atmosphere, duration, temperature, and the inclusion of special additives, it becomes possible to adjust both the total specific surface area and the internal porous structure of the resulting PCM, as well as its conductive properties [36,37]. In [36], it is reported that increasing the carbonization temperature of biomass from 400 °C to 1000 °C leads to a decrease in the specific resistance of the obtained carbon material by more than eight orders of magnitude.

An analysis of the literature sources shows that researchers pay insufficient attention to the study of simple and affordable methods for obtaining and activating carbon material. The study aimed to investigate the effect of the activation duration in an atmosphere with limited air access on the structure, morphology, and electrochemical properties of the PCM. To achieve this goal, the activation of the starting carbon material was carried out and the effect of its duration on the specific surface area, pore size distribution, fractal dimension, average size of graphite fragments, and energy characteristics of PCM in 6 M KOH was investigated.

## 2. Materials and Methods

To obtain carbonized carbon material, the original bio-raw material (walnut shells) was introduced into an autoclave without additional grinding. The autoclave was then placed in a furnace and heated at a rate of 10 °C/min to a temperature of 800 °C, where it was held for 1 h. The work uses an autoclave designed for hydrothermal synthesis, which is made of stainless steel and can withstand 800 °C degrees. Carbonization is carried out in an atmosphere of residual gases, that is, in an atmosphere containing CO_2_, which is formed during the thermal decomposition of biomass. Afterwards, the furnace was turned off and allowed to cool to room temperature. The resulting carbonized material was then crushed using mechanical grinding, resulting in the production of carbon material, designated as CC [38].

The effectiveness of the process of grinding raw materials depends on understanding the mechanisms of fragmentation of shell structures, in particular, by brittle fracture of the material near closable defects [39,40,41,42,43,44,45]. In particular, article [39] considers the problem of shelling a mass of nuts under the press plate. To preserve the integrity of the nut grains, the conditions for the controlled sustainable development of cracks are defined here. Mechanisms of catastrophic propagation of contact cracks from the standpoint of the energy criterion of brittle fracture mechanics were studied in works [40,41,44] for cylindrical shells, and in articles [42,43] for shells of double curvature. The effect of interference of closable defects on the destruction of shell structures was investigated in [45].

The carbon material (CC) was placed into a ceramic vessel and subjected to additional activation in a muffle furnace. Thermal activation of this carbon was conducted at a temperature of 400 °C, with activation durations of 1 h designated as CCA1, 2 h as CCA2, and 3 h as CCA3. Figure 1 shows the process scheme for obtaining PCM from walnut shells. 

For carbonation, a walnut shell weighing 50 g was placed in the autoclave, the weight of the PCM output was ~20 g, depending on the carbonation temperature. A total of 2 g of PCM was taken for activation. The yield of the final product at 1 h of activation was about 85% (~1.7 g), at 2 h—about 70% (~1.4 g), at 3 h—50% (~1 g). This was also one of the reasons why the activation duration was no longer increased.

X-ray patterns of carbon samples were obtained using XRD-7000, Shimadzu diffractometer (Shimadzu Corporation, Kyoto, Japan), equipped with a graphite monochromator (wavelength of 1.5405 Å for Cu–K_α_ radiation). Small-angle X-ray scattering (SAXS) patterns of carbon samples were obtained on Shimadzu XRD-7000S operated at 40 kV and 30 mA.

The specific surface area (S_BET_), pore size distribution, and fractal dimension (*D*) of carbon samples were determined by the method of low-temperature nitrogen adsorption on an Autosorb Nova 2200e analyzer (Quantachrome Instruments, USA).

Nitrogen adsorption/desorption isotherms were obtained in the range of relative pressures *P*/*P*_0_ from 0.0 to 1.0 at a temperature of −196 °C.

The value of S_BET_ was determined by the Brunauer–Emmett–Teller multipoint method (MultiPoint BET) [46]; to determine the surface area of micropores (Smic), the Halsey t-method (according to the de Boer equation) was used [47]; and the total pore volume (*V*_∑_) was calculated from the volume of adsorbed nitrogen at *P*/*P*_0_ ≈ 1. The volume of mesopores (*V*_meso_) and their size distribution were calculated using the Barrett–Joyner–Halenda method (BJH method), while the density functional theory (DFT method) was used for the distribution of micropores [48]. The fractal dimension (*D*) was calculated using the Frenkel–Halsey–Hill method (FHH-method) [49,50]. All parameters of the porous structure of PCM were calculated using Quantachrome NOVA 2200e software.

The Raman spectra were recorded over a Raman shift range from 100 to 4500 cm^−1^ using a T64000 Jobin Yvon spectrometer (Horiba, Kyoto, Japan) (1800/mm, resolution about 1 cm^−1^) in reverse dispersion geometry using an argon–krypton laser (*λ* = 488 nm). The power of laser irradiation, <1 mW/cm^2^, allows for avoiding the local overheating of the sample surface.

The morphology of the investigated porous carbon materials was studied using a JSM-6700F field-emission scanning electron microscope (JEOL, Akishima, Japan). Secondary electron images were obtained on carbon-coated samples under operating conditions of an accelerating voltage of 10 kV and a beam current of 0.75 nA.

Electrochemical investigations were performed on an Autolab PGSTAT/FRA-2 spectrometer (Autolab, Amsterdam, The Netherlands) using the methods of cyclic voltammetry, galvanostatic cycling, and impedance spectroscopy. All measurements were carried out in a two-electrode cell. The carbon electrodes were made in the form of lamellae using a mixture consisting of 75% PCM and 25% conductive additive. The research was conducted in a 6 M aqueous KOH solution with a maximum charge/discharge voltage set at 1 V. Distilled water and KOH salt (chemically pure) were used to make the electrolyte.

## 3. Results and Discussion

The structural arrangement of porous carbon is intermediate between crystalline amorphous materials and can be described as a hierarchical system composed of turbostratic nanodomains formed by distorted stacked packages of graphitic layers randomly oriented and connected by *sp*^3^ bonds. This model implies the presence of both micro- (size less than 2 nm) and mesopores (size in a range of 2–50 nm) whose type (cylindrical, slit-like, etc.) and concentration will define the adsorption and capacitive properties of the material. All synthesized carbon samples show similar diffraction patterns with two broad peaks at 2*θ* angles of about 23° and 43° corresponding to the (002) and (10) reflections of a turbostratic carbon. The (10) peak is a result of the overlapping (100) and (101) reflections and its presence in the neighborhood of the graphite (100) reflexes. Typically this peak is observed for highly defect porous carbons with the presence of disordered stacking graphitic layers. In research articles, this peak typically is named as «two-dimensional (10) reflection» [51]. The activation duration growth causes the increases in the full-width half-maximum (FWHM). The obtained XRD patterns were analyzed using Match! 3.12 with pseudo-Voigt functions for the determination of (002) and (10) peaks angular positions (2*θ*_hkl_, in radians) and FWHM values (*β*_hkl_, in radians). The average interplanar spacing *d*_002_ values were calculated according to Bragg’s law: *d*_002_ = *λ*/(2*sinθ*_002_). The graphitic crystallite size (*D*_002_) values along the perpendicular to basal (002) plane were estimated Debye–Scherrer equation, *D*_002_ = 0.89*λ*/*β*_002_*cosθ*_002_, when lateral sizes of graphitic crystallites (*L*_XRD_) along the basal (002) plane were calculated as *L*_XRD_ = 1.84*λ*/*β*_10_*cosθ*_10_ [52]. The increase in carbonization duration causes a decrease in interplanar spacing (Figure 2a). The observed values of interplanar spacing (*d*_002_) are in a range of 0.38–0.36 nm that corresponds to the distortion of the graphitic lattice comparatively to graphite with *d*_002_ = 0.335 nm. The noticeable decrease in *D*_002_ crystallite size is observed only for activation time of 3 h (Figure 2b), which corresponds to reducing the average number of stacked layers in the graphitic domains.

The decrease in lateral size *L*_XRD_ of graphitic crystallites from 3.79 to 2.52 nm reflects the changes in *d*_002_ (correlation coefficient between *d*_002_ and *L*_XRD_ equals *r* = 0.88) (Figure 2c). Thus, the increase in the activation duration at fixed temperature causes a decrease in both the thickness and lateral sizes of graphitic crystallite.

According to low-temperature porosimetry data, PCM is characterized by nitrogen adsorption/desorption isotherms of type IV according to the IUPAC classification [53], both for the original CC sample (Figure 3a) and for thermally activated samples CCA1–CCA3 (Figure 3b,d). As can be seen from the figures, there are hysteresis loops of high and low pressure on the isotherms, indicating the presence of a significant number of mesopores 2–50 nm in size in the materials, where irreversible capillary condensation occurs. However, on the isotherms of samples CC and CCA1 (Figure 3a,b), there is a hysteresis of the H4 type (according to IUPAC), which is characteristic of materials with narrow slit-like pores or complex materials containing micro- and mesopores. For samples CCA2 and CCA3, the isotherms exhibit H3-type hysteresis loops (Figure 3c,d). This type of hysteresis is commonly associated with materials featuring slit-like pores, although the presence of pores with different shapes, such as cylindrical ones, cannot be ruled out.

The obtained isotherms (Figure 3) enabled the calculation of the porous structure parameters of PCM and facilitated the analysis of the impact of thermal activation on the specific surface area and pore size distribution. The characteristics of the porous structure of PCM before and after activation are presented in Table 1.

As observed in Table 1, thermal activation at a temperature of 400 °C results in the development of a porous structure in the carbon material, irrespective of the activation time. The specific BET surface area for activated samples increases by approximately two times. The *S*_BET_ value was determined based on the adsorption isotherm within the range of relative pressures from *P*/*P*_0_ = 0.05 to 0.30. PCM with a maximum specific surface area of 431 m^2^/g and a pore volume of 0.521 cm^3^/g was obtained with an activation duration of 2 h. The fact that there is no big difference in the surface area due to the different duration of activation may be because if the carbon material has a well-developed porous structure (as far as it is possible for this method of obtaining PCM) already after 1 h of thermal activation, then the duration of thermal pre-activation may have a minor effect on the specific surface area of the material. Heat treatment can stabilize the pores of the carbon material, preventing their destruction [54,55]. Analysis of the porosimetry data (Table 1) suggests that thermal activation resulted in the formation of mesoporous carbon materials, characterized by a relative content of mesopores of approximately 75–78% and an average pore diameter of about 5 nm.

The size distribution of mesopores for the PCM samples was calculated using the BJH method, considering relative pressures *P*/*P*_0_ greater than 0.35 (Figure 4). The graph shows several maxima, indicating the presence of multiple groups of pores in the samples, each characterized by different average diameters, *d*. In the mesopore region of the initial sample (CC), peaks are observed at diameter values of 3.2, 4.0, 6.2, and 8.8 nm (Figure 4a), indicating a polymodal structure in the samples. When comparing the distribution curves for activated samples (CCA1–CCA3) with the original sample, it is evident that there is an increase in the peak for pores with an effective diameter of 4 nm in the activated samples, indicating an increase in the number of pores of the appropriate size (Figure 4b,d).

Since the BJH method is limited to relative pressures *P*/*P*_0_ < 0.35, nonlocal density functional theory (NL-DFT) for the slit-pore model was employed to analyze the distribution of micropores at *P*/*P*_0_ > 0.15 (Figure 5). This method allows for the determination of pore size distributions ranging from 0.35 to 40 nm in average diameter. One notable feature of the NL-DFT method is its capability to describe the gradual change in adsorbate density near the pore wall. This stands in contrast to classical theories like Kelvin’s equation, which forms the basis of the BJH method and assumes a uniform density similar to the liquid state.

The pore distribution obtained by the DFT method indicates that the original PCM (Figure 5a) is a microporous material characterized by pores in the range of 1.5–2 nm. The relative volume content of mesopores in the CC sample is reported as 10.5% in Table 1. Thermal activation promotes the development of the micro- and mesoporous structure in the carbon material. In the carbon material activated for 1 h (Figure 5b), a decrease in the volume of micropores is observed from 0.063 cm^3^/g to 0.043 cm^3^/g, while activation for 2 and 3 h (Figure 5c,d) is accompanied by an increase in the corresponding value to approximately 0.08 cm^3^/g. The distribution of pores demonstrates that for activated samples CCA1–CCA3, the main pore volume shifts within the ranges of micropores 1.5–2 nm and mesopores 3–8 nm ± 0.15 nm, and this shift appears to be relatively independent of the duration of activation.

One of the most effective methods to characterize the irregular and complex pore structure of carbon materials is the Frenkel–Halsey–Hill method [56,57]. According to this method, the fractal dimension can be calculated using the following equation based on nitrogen adsorption data [58]:$$\ln\left(\frac{V}{V_{0}}\right)=C+A\left[\ln\left(\ln\left(\frac{P_{0}}{P}\right)\right)\right],$$
A=D−3,
where *V* represents the volume of nitrogen adsorbed by PCM at equilibrium pressure; *V*_0_ is the volume of nitrogen adsorbed by the PCM monolayer at saturation pressure; *P*_0_ denotes the pressure of saturated nitrogen vapor; *A* is a power indicator dependent on the fractal dimension and adsorption mechanism; *C* is the gas adsorption constant; and *D* represents the fractal dimension.

The fractal dimension is used for estimating zeros of Riemann’s zeta function [59] and other special functions [60]. The fractal dimension of the PCM surface before and after thermal activation was studied by analyzing the graphical dependence ln*V* from ln[ln(*P*_0_/*P*)] (Figure 6). The fractal dimension was calculated using nitrogen adsorption isotherms (Figure 3), with the effects of adsorbate surface tension being neglected. From Figure 3, it can be inferred that the isotherms exhibit slow growth in the region of low relative pressure (*P*/*P*_0_ ≤ 0.4), which is attributed to monolayer adsorption controlled by the Van der Waals forces prevalent in this region. In the region of high relative pressure (*P*/*P*_0_ > 0.4), multilayer adsorption occurs, which is primarily controlled by surface tension [61,62]. Considering the differences in gas adsorption mechanisms between low- and high-pressure regions, the fractal dimensions of the original and activated PCM were designated as *D_F_*_1_ and *D_F_*_2_, respectively. The approximation curves were segmented at the demarcation point *P*/*P*_0_ ≈ 0.4, indicating the presence of two distinct fractal features in the pores of the investigated PCM. Hence, the presence of two clear ranges of pores, micro and meso, was evidenced by the two distinct linear plots observed in the dependence of ln*V* on ln[ln(*P*/*P*_0_)] for all PCM samples.

The results of the study indicate that the values of *D*_*F*1_ and *D*_*F*2_ for PCM are between 2 and 3, as shown in Table 2, and stand together with the definition of the fractal dimension for carbon materials.

The *D*_*F*1_ values for thermally activated samples CA1–CA3 are significantly higher (2.52, 2.69) than for the original CA sample (2.17), whereas the *D_F_*_2_ values for the original sample are higher than for the activated samples, which are 2.88 and approximately 2.6, respectively. All fitting curves exhibited excellent fits, with *R*^2^ correlation coefficient values exceeding 0.99.

As noted in [49], *D_F_*_1_ represents the surface roughness of micropores. As the time of thermal activation increases, the values of *D_F_*_1_ (Table 2) increase, indicating that the surface of micropores in activated PCM samples becomes rougher. Therefore, *D_F_*_2_ can be used to describe the fractal characteristics of the mesopore volume.

For activated PCM samples, the value of *D_F_*_2_ (Table 2) increases with activation durations of 1 and 2 h, reaching a maximum value of 2.62. This indicates that the internal structure of mesopores became more complex during activation over this time, implying an increase in the number of mesopores. Activation for 3 h, however, does not alter the *D_F_*_2_ parameter.

For the original CC sample, the value of *D_F_*_1_ = 2.17 is notably lower than the value of *D_F_*_2_ = 2.88. This suggests that the surface of the micropores in this carbon material is smoother, while the surface of the mesopores is much rougher.

Numerous experiments have shown that the Raman spectra of PCM are very sensitive to the organization of their structure [63,64,65]. The Raman spectroscopy method allows for obtaining information about the size of atomically organized regions in carbon nanomaterials, the structure, type of mutual ordering, atomic geometry of the boundaries of such formations, and many other important characteristics of PCM. Analyzing both the qualitative and quantitative attributes of the Raman spectrum obtained, it becomes feasible to ascertain the existence of graphite and graphene in the samples under investigation, along with evaluating their defects and turbostratic alterations. Raman spectroscopy exhibits its highest sensitivity towards highly symmetric covalent bonds possessing minimal or zero dipole moment, and carbon bonds perfectly fulfil this criterion [64]. Hence, this method enables the detection of even subtle alterations in the structure of PCM.

Figure 7 shows the Raman spectra (RS) of both the initial and activated PVM. Within these spectra, two bands (*D* and *G*) are evident, characteristic of the first-order Raman scattering of disordered amorphous carbon [38]. The *G* (“Graphite”) band, appearing around ~1580 cm^−1^, originates from the active Raman mode E_2g_ of graphite. The *D* (“Defect”) band, occurring at approximately ~1340 cm^−1^, represents a band of disordered carbon structure induced by the vibrational mode of the graphite lattice with A_1g_ symmetry [66]. All samples exhibit a higher integrated intensity of the *D*-band and broad *G*-bands, suggesting their low-ordered structure. The ratio of the intensity of the *D* and *G* bands can serve as a measure to estimate the degree of disorder in the carbon material [67]. The width of the *D*- and *G*-bands was determined based on two Lorentzians (Figure 7), using the OriginPro 8.5 program. Fittings with a coefficient of determination *R*^2^ > 0.96 were achieved for the analyzed PCM samples. The results obtained indicate that for the activated samples CCA1–CCA3, the ratio of integral intensities *I_D_*/*I_G_* is notably higher compared to the initial sample CC. This observation can be attributed to an increase in the amorphous component and a rise in the porosity of the carbon material following thermal activation, which agrees with the findings of low-temperature porosimetry.

Analyzing the integral intensities of the *D*- and *G*-bands (*I_D_* and *I_G_*, respectively) enables estimation of the average size *L* of graphite fragments along the basal plane (002) of graphite [68]:$$L\left(\mathrm{nm}\right)=\left(2.4\times 10^{-10}\right)\lambda^{4}\left(\frac{I_{G}}{I_{D}}\right),$$
where *λ* = 488 nm is the wavelength of laser radiation.

The values of the average size of graphite fragments for activated PCM samples are significantly smaller compared to the original sample, as illustrated in Figure 8. The average transverse particle size of graphite fragments undergoes a change from approximately ~8 nm to ~5 nm during the activation process. It can be assumed that this phenomenon is caused by simultaneous or sequential processes. One of the processes involved is the thermal decomposition of organic compounds, which takes place as the temperature rises or persists during further heat treatment (activation) of PCM. The material undergoes degradation of its carbon structures as a result of temperature exposure, resulting in the generation of smaller graphite fragments. Furthermore, throughout the heat treatment of carbon material, both evaporation and decomposition of surface groups transpire, resulting in an enlargement of pores within the material and a reduction in graphite fragments.

Figure 9 shows the SEM images of the surface morphology of the investigated PCM. As depicted in Figure 9, both the initial sample (CC) and the thermally activated (CCA2) exhibit porous characteristics with pores of varying diameters. The carbon particles of the thermally activated material appear to be more porous and rougher, with smaller carbon particle fragments observed on their surface. From a visual perspective, one might hypothesize that the pores within carbon particles resemble a root-like structure, and thermal activation results in their expansion and branching.

Thermal activation results in the formation of surface functional groups. The identification of these functional groups on carbon materials was conducted by analyzing the IR spectra of carbonized and activated PCM (see Figure 10).

The IR spectra of both carbonized and thermally activated PCM samples exhibit absorption bands in the range of 3400–3500 cm^−1^. These bands are attributed to the valence O–H vibrations of phenol and alcohol groups. The peak at 3450 cm^−1^ is probably due to fluctuations in the O–H bond of physically adsorbed water. The IR spectra of all carbon samples exhibit modes at 2850 and 2925 cm^−1^, corresponding to symmetric and asymmetric valence C–H vibrations in the CH_2_ groups. The presence of O–H groups on the surface of the investigated PCM is verified by the peak observed at 1635 cm^−1^. After thermal activation, a peak at 1455 cm^−1^ is observed, indicating the presence of deformation vibrations of CH_2_ groups. It should be noted that increasing the duration of activation results in a corresponding increase in the intensity of the data. The peaks in the range of 700–900 cm^−1^ originate from C–H vibrations of phenolic groups. The peak at 940 cm^−1^ indicates the presence of carboxylic acid residues on the PCM surface, while the peak at 1115 cm^−1^ suggests the presence of ozonides.

To obtain information occurring in the electrochemical system, the method of cyclic voltammetry (CVA) was used in the range of scan rate from 1 to 50 mV/s (Figure 11a,b). Changing the scan rate of the potential allows for the acquisition of insights into the charge accumulation mechanisms within the electrochemical system. The shape of the CVA curves (Figure 11a,b) indicates the predominantly capacitive nature of charge accumulation, as they exhibit a relatively symmetrical shape without redox peaks. This indicates the quasi-reversibility of the charge/discharge processes within the investigated electrochemical systems [38]. However, more pronounced deviations from the practically rectangular shape are observed on the CVA curve for the CCA2 sample compared to the other samples. This observation indicates the pseudocapacitive properties of this PCM (Figure 11a) [69]. This is most likely due to the formation of oxygen functional groups on its surface during the thermal activation process, with 2 h of activation being the optimal time interval for this process. At a scan rate of more than 20 mV/s, the curves deviate from a rectangular shape (Figure 11b), but the ability to reverse cycling remains, indicating the potential use of these materials as electrodes for electrochemical capacitors.

The specific capacity (*C_sp_*) of the investigated PCM was calculated using the ratio of $C_{sp}=2q/{(mU}_{max})$, where *q* represents the charge in coulombs, *U*_max_ is the maximum charge/discharge voltage in volts, and m is the mass of the electrode in grams. The results show that the specific capacitance of the PCM monotonically decreases with the increasing scan rate (Figure 11c). This is explained by the fact that as the scanning speed (*s*) increases, the number of pores inaccessible to the electrolyte (mainly ultra- and micropores) also increases. A sharp decrease in capacity with an increase in the scanning speed also occurs as a result of an increase in the internal resistance of the transfer of electrolyte ions. Consequently, a smaller surface area of the PCM will participate in the formation of the electric double layer. It should be noted that thermal activation for 2 h contributes to an increase in the specific capacity of the corresponding electrochemical system. This is likely due to the material’s largest specific surface area and optimal structural parameters.

A visual analysis of the CVA curves (Figure 11a,b) suggests that in the EC based on the obtained PCM, in addition to the EDL capacity (*C_EDL_*), there is also a pseudo-capacitance (*C_F_*) contributing to the total capacity (*C*). The authors of [70] proposed a method that allows for the separation of contributions from different charge accumulation mechanisms (capacitance accumulation) to the total capacity of PCM: $C=C_{DEL}+C_{F}$, where *C_F_* represents the diffusion-controlled redox capacitance, also known as pseudo-capacitance. This pseudo-capacitance arises from Faraday reversible redox reactions due to functional surface groups. As stated in [70], extrapolating the dependencies of *C* from $1/\sqrt{s}$ to the *Y* axis (Figure 12a) enables the calculation of the specific capacity, attributed to the formation of a double electric layer at the carbon electrode/electrolyte interface (Table 3). Similarly, as per [70], the reciprocal of *C* will exhibit a linear dependence on $\sqrt{s}$ (Figure 12b), and extrapolating this relationship to the Y-axis enables the determination of the maximum specific capacity, *C*, of the carbon materials obtained (Table 3).

Figure 13a shows typical galvanostatic charge/discharge curves for an electrochemical system made based on carbon material CCA2, the appearance of which confirms the predominant capacitive mechanism of charge accumulation in the electric double layer. The galvanostatic charge/discharge curves for the other samples exhibit a similar shape and are, therefore, not included in the paper. The specific capacity *C_sp_* (F/g) of the PCM was calculated at a charge/discharge current of 2, 5, 10, 20, 30, 40, and 50 mA, according to the ratio $C_{sp}=2It/{[(U}_{\max}-∆U)m]$, where *I* is the discharge current in *A*, *t* is the discharge time in seconds, (*U*_max_ − Δ*U*) is the operating voltage window in V, Δ*U* is the voltage drop when the discharge circuit is closed, and m is the mass of the PCM. Figure 13b shows the relationship between the specific capacity of the carbon material and the value of the charge/discharge current. It is determined that the CCA2 sample exhibits a higher specific capacity (~110–130 F/g) across all values of discharge currents.

The effect of thermal activation and its duration on the electrochemical characteristics of capacitors constructed based on the obtained PCM was also studied using impedance spectroscopy. Figure 14a displays the resulting Nyquist diagrams in the frequency range of 0.01–100 kHz. In the high-frequency range, a small semicircular area is observed, which is associated with the resistance of charge transfer in the porous structure of the carbon electrode [71]. In the medium frequency range, a linear sloping section is observed, which is associated with diffusion processes occurring in the mesopores of the carbon electrodes [72]. At low frequencies, an almost vertical line is observed on the experimental impedance spectra, indicating the capacitive mechanisms of charge accumulation in the narrow mesopores and micropores of the electrodes. It is worth noting that the maximum values of specific capacitance are typically attained at very low frequencies. Therefore, electrochemical capacitors are ideally applicable at such low frequencies or under constant current conditions [73].

The experimentally obtained Nyquist diagrams were described using the electrical equivalent circuit (EES) (Figure 14b). The presented EES includes the following components: inductance (*L*), which is caused by the presence of conductors and lead contacts; series equivalent resistance (*R*_s_), which represents the resistance of the electrolyte, electrode material, and resistance caused by the cell’s design; link CPE_1_║*R*_1_, reflecting the diffusion processes in the macropores of the porous electrode and the processes of charge accumulation on their surface; and link CPE_2_║*R*_2_–C, which is responsible for charge accumulation processes in meso- and micropores. In the proposed scheme (Figure 14b), CPE elements are used to provide flexibility in modeling. Specifically, CPE_1_ is a constant phase element of capacitive type for activated samples CCA1–CCA3 and diffusion type for the carbonized sample CC. The constant phase element of the diffusion type CPE_2_, is associated with the processes of limited diffusion of K^+^ and OH^−^ ions in micropores. Resistances *R*_1_ and *R*_2_ are resistances for charge transfer in the pores of the carbon material, and *C* is the capacity accumulated in meso- and micropores.

The proposed EES (Figure 14b) is well-suited for approximating the experimental spectra. The values of the parameters included in the scheme are obtained using the EES (Table 4). Analysis of the parameter values reveals the CCA2 sample as exhibiting the highest specific capacity values, consistent with previous studies. At the same time, this sample demonstrates the smallest of solute and charge transfer resistances. Most likely, this is related to the parameters of the porous structure of this sample. Typically, carbon materials with a larger surface area tend to have lower solution resistance. This is because a larger surface area provides shorter paths for the diffusion of electrolyte molecules through the electrode.

Hence, a thermal activation duration of 2 h proves optimal for achieving PCM with ideal structural and morphological characteristics for electrochemical capacitor electrodes. In a 6 M KOH electrolyte, this process yields a specific capacity of approximately 125 F/g at a discharge current density of 10 mA/g.

In further research, it is planned to use other raw materials of natural origin for this activation method and to expand the activation time interval. The obtained PCMs will be investigated as electrode materials for electrochemical power sources.

## 4. Conclusions

Porous carbon material was obtained from walnut shells through thermal carbonization at 800 °C followed by subsequent thermal activation at 400 °C for 1, 2, and 3 h. The morphology and structure of the obtained carbon materials were researched using SAXS, SEM, Raman spectroscopy, and low-temperature porometry. Results demonstrate that increasing the activation duration at a constant temperature causes a reduction in both the thickness and lateral dimensions of graphite crystallites. The average sizes of graphite fragments in the obtained carbon materials were determined, revealing that thermal activation at 400 °C reduces the average cross-sectional size of graphite particles from approximately 8 nm (for carbonized PCM) to around 5 nm (for activated PCM). Thermal activation is shown to enhance the development of a mesoporous structure in the PCM, with a relative mesopore content of approximately 75–78% and an average pore diameter of around 5 nm. This process approximately doubles the specific surface area of the original PCM. The fractal dimension of the obtained carbon materials in the region of low (*P*/*P*_0_ ≤ 0.4) and high (*P*/*P*_0_ > 0.4) relative pressures was calculated using the Frenkel–Halsey–Hill method.

The obtained carbon materials were used to manufacture electrodes of electrochemical capacitors based on a 6 M KOH electrolyte. The electrochemical behavior of these systems was studied by the CVA method, galvanostatic cycling, and SEI, and specific capacitance values for these systems were obtained. It was determined that the carbon material activated at 400 °C for 2 h is characterized by a specific capacity of ~130–110 F/g at a discharge current density of 4–100 mA/g and a maximum charge voltage of 1 V. Based on the SEI method, the EES was proposed and the physical interpretation of the circuit elements was presented.

## Figures and Tables

**Figure 1 materials-17-02514-f001:**
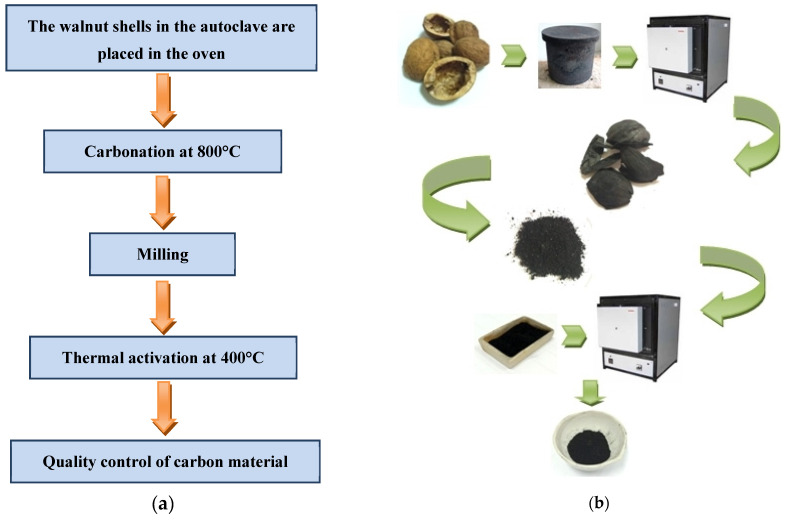
A list of the main operations of the technological process (**a**) and a schematic illustration of obtaining PCM from walnut shells (**b**).

**Figure 2 materials-17-02514-f002:**
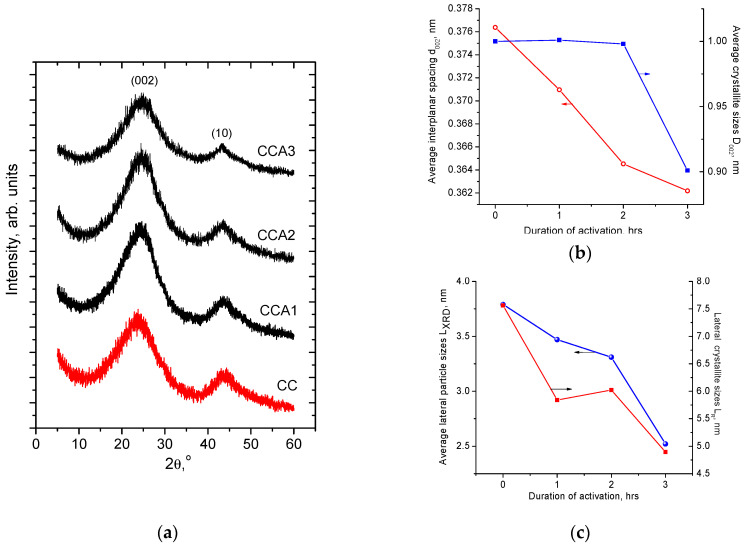
XRD patterns of the carbon samples obtained at the different duration of activation process (**a**), interplanar spacing *d*_002_ (red line) and crystallites thickness *D*_002_ (blue line) (**b**), lateral crystallites sizes *L*_XRD_ (blue line) and *L*_R_ (red line) from XRD and Raman data, respectively (**c**).

**Figure 3 materials-17-02514-f003:**
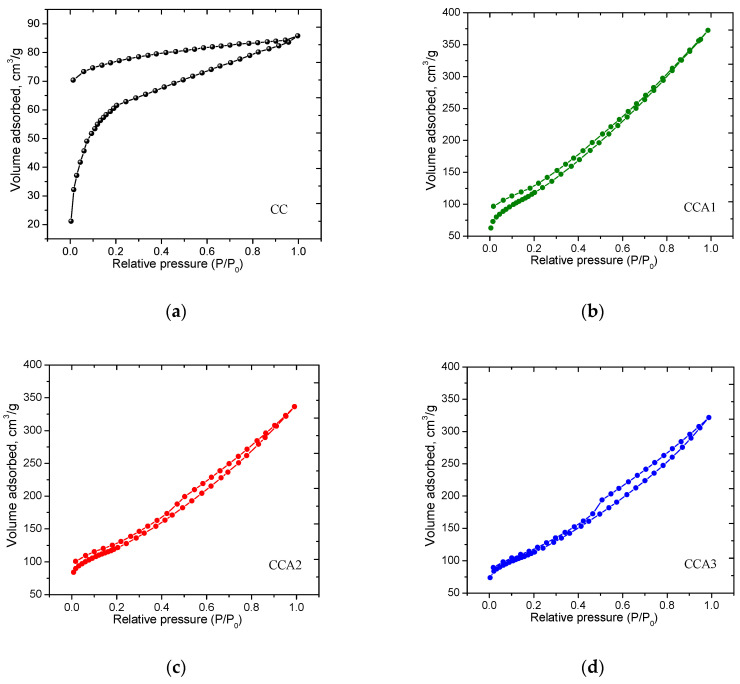
Nitrogen adsorption/desorption isotherms for carbonized—sample CC (**a**) and activated PCM with activation times of 1 h—sample CCA1 (**b**), 2 h—sample CCA2 (**c**), and 3 h—sample CCA3 (**d**).

**Figure 4 materials-17-02514-f004:**
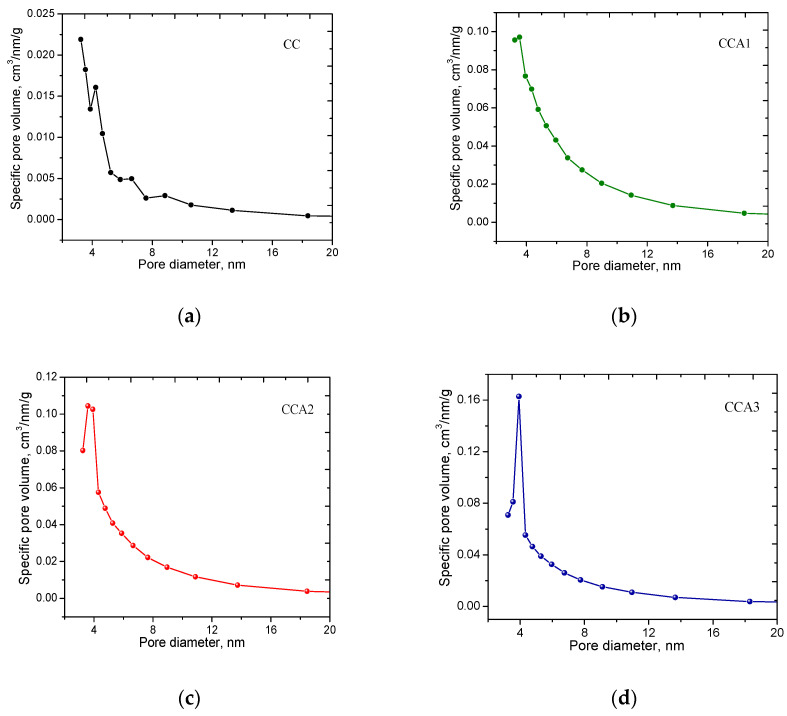
Distribution of mesopores by size according to the BJH method for PCM samples: (**a**) initial sample CC and activated samples with activation durations of 1 h—sample CCA1 (**b**), 2 h—sample CCA2 (**c**), and 3 h—sample CCA3 (**d**).

**Figure 5 materials-17-02514-f005:**
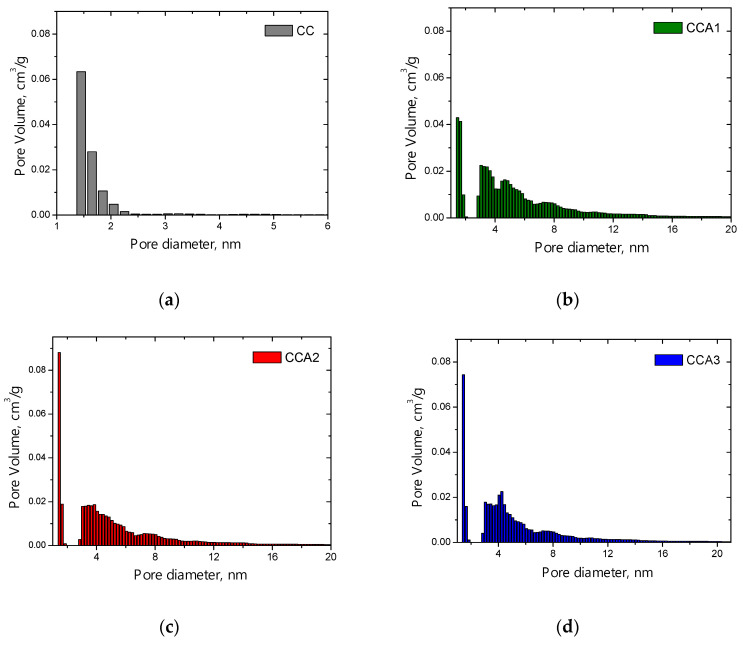
Size distribution of micropores for the initial PCM sample (**a**) and activated PCM samples with different activation durations (**b**–**d**), calculated using the DFT method.

**Figure 6 materials-17-02514-f006:**
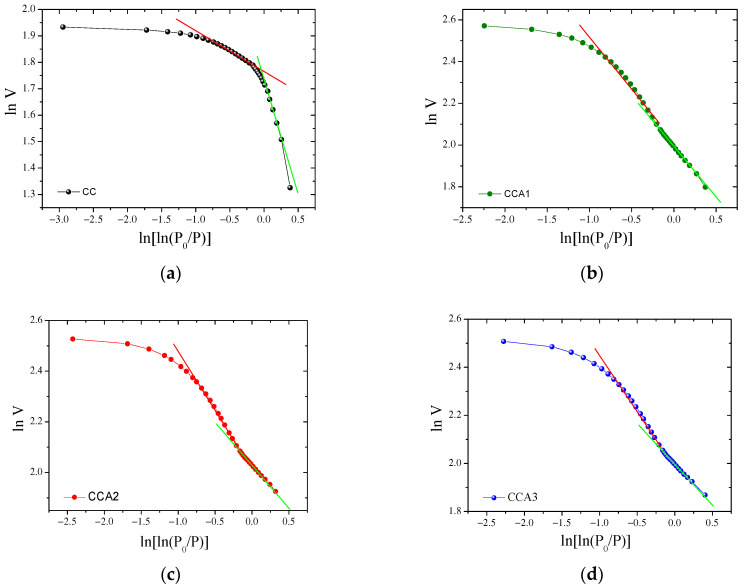
Graphical dependencies of ln*V* on ln[ln(*P*_0_/*P*)] for PCM samples: (**a**) initial sample CC and activated samples with activation durations of 1 h—sample CCA1 (**b**), 2 h—sample CCA2 (**c**), and 3 h—sample CCA3 (**d**). Approximation curves for low-pressure region—green line and the high-pressure region—red line.

**Figure 7 materials-17-02514-f007:**
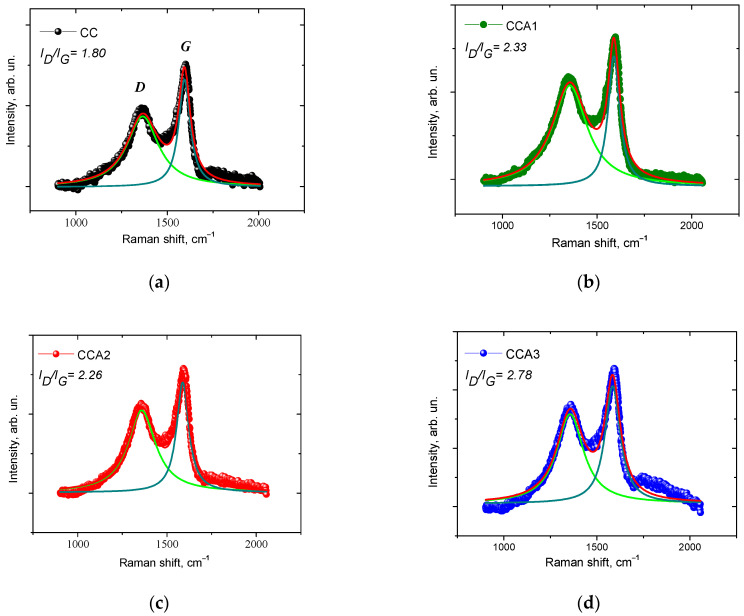
Raman spectra for initial (**a**) and thermally activated (**b**–**d**) PCM samples. Lorentzians for determining the width of the *D*-band—green line and *G*-band—blue line.

**Figure 8 materials-17-02514-f008:**
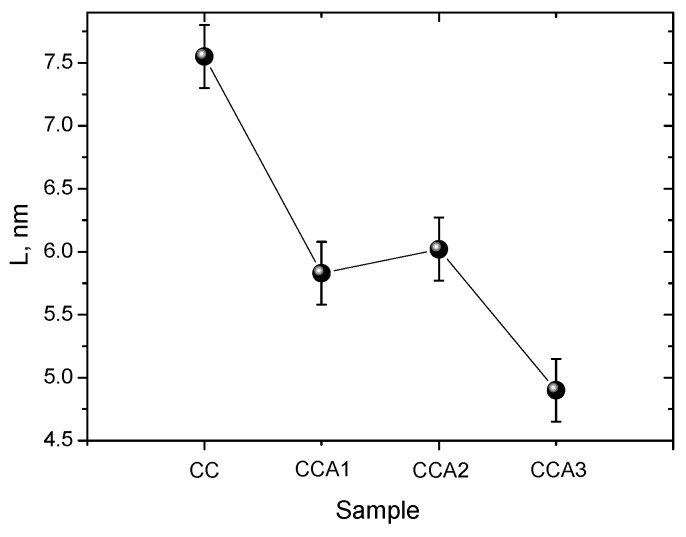
The average size of graphite fragments for PCM samples.

**Figure 9 materials-17-02514-f009:**
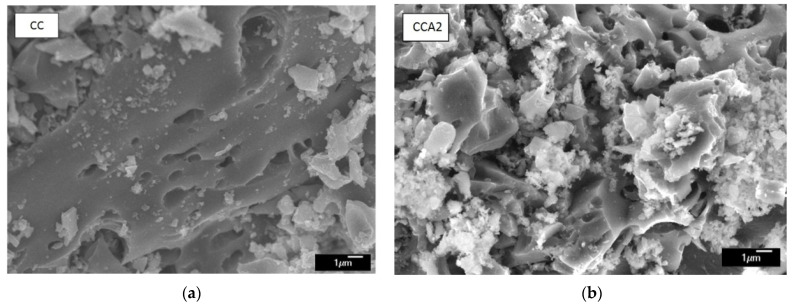
SEM image of carbonized (**a**) and activated at 400 °C for 2 h (**b**) carbon material.

**Figure 10 materials-17-02514-f010:**
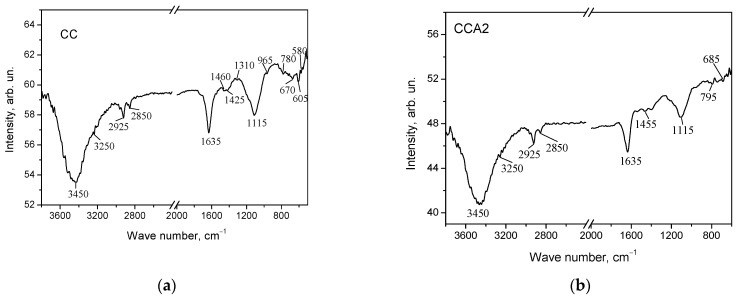
IR spectra of the surface of carbonized (**a**) and activated at 400 °C for 2 h (**b**) carbon materials.

**Figure 11 materials-17-02514-f011:**
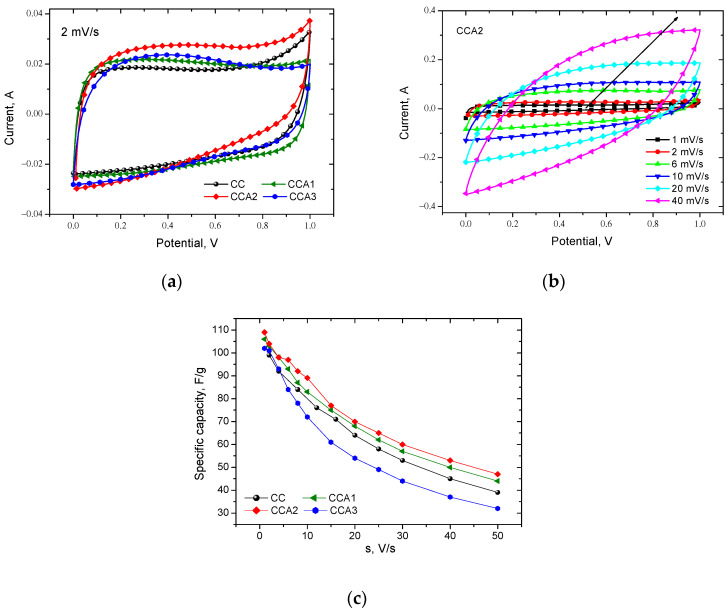
CVA curves for PCM samples in 6 M KOH (**a**,**b**) and dependences of the specific capacitance of PCM on the scan rate, calculated from the data of CVA (**c**).

**Figure 12 materials-17-02514-f012:**
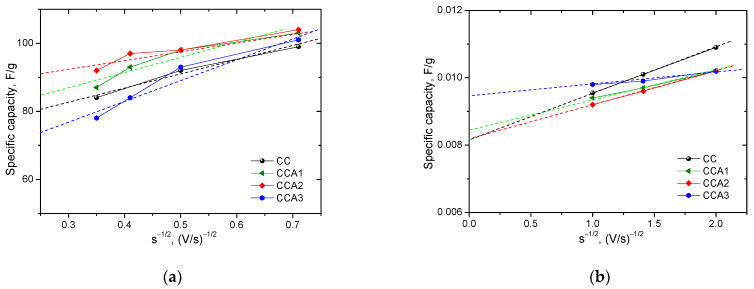
Dependencies of *C* on $1/\sqrt{s}$ (**a**) and 1/*C* on √*s* (**b**) for PCM samples (Approximation lines–dashed lines.).

**Figure 13 materials-17-02514-f013:**
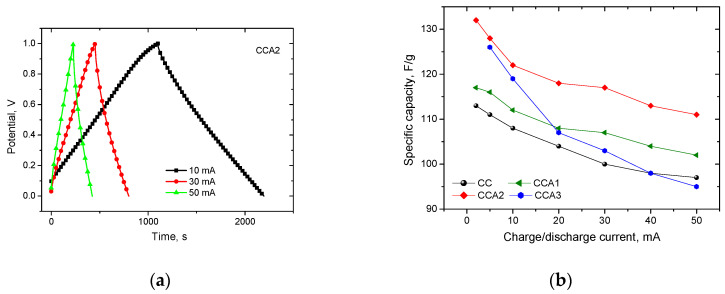
Typical charge/discharge curves for CCA2 in 6 M KOH (**a**) and the dependence of the specific capacity of the studied samples of PCM on the magnitude of the discharge current (**b**).

**Figure 14 materials-17-02514-f014:**
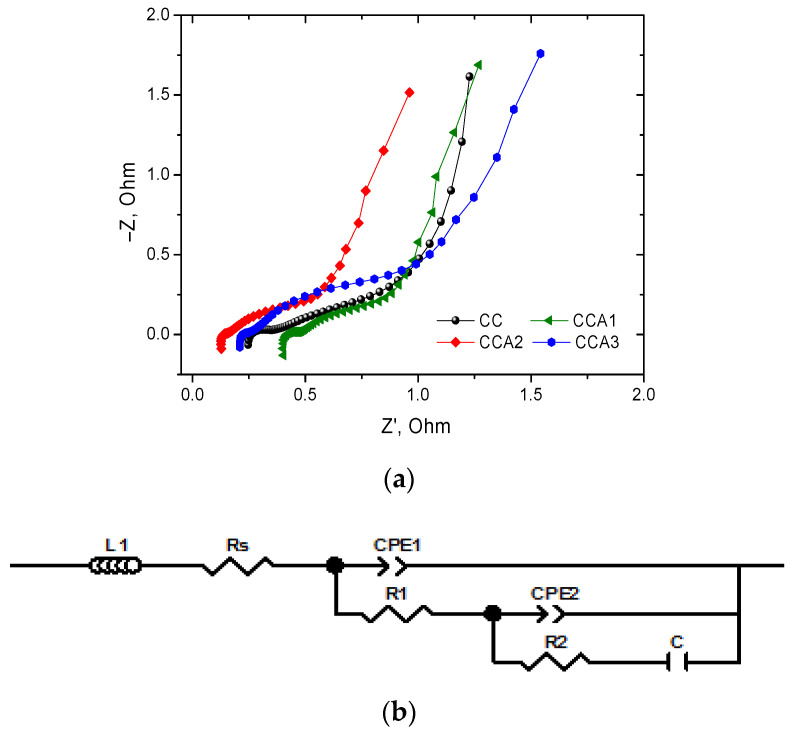
Nyquist diagram for the electrochemical capacitor at an open circle voltage (**a**), EEC for approximate the impedance spectra (**b**).

**Table 1 materials-17-02514-t001:** Parameters of the porous structure of PCM.

Sample	Surface Area, m^2^/g		Pore Volume, cm^3^/g		*V*_meso_/*V*_∑_, %	*d*_por_, nm
	*S* _BET_	*S* _micro_	*V* _∑_	*V* _meso_		
CC	238	133	0.133	0.014	10.5	2.2
CCA1	421	–	0.576	0.445	77.3	5.5
CCA2	431	–	0.521	0.390	74.9	4.8
CCA3	400	–	0.498	0.387	77.7	5.0

**Table 2 materials-17-02514-t002:** The surface fractal dimensions were calculated using the FHH fractal model.

Sample	Low-Pressure Region (*P*/*P*_0_ ≤ 0.4)			High-Pressure Region (*P*/*P*_0_ > 0.4)		
	Slope	*R* ^2^	*D_F_* _1_	Slope	*R* ^2^	*D_F_* _2_
CC	−0.830 ± 0.003	0.996	2.17	−0.115 ± 0.003	0.991	2.88
CCA1	−0.479 ± 0.002	1.0	2.52	−0.418 ± 0.02	0.991	2.58
CCA2	−0.308 ± 0.002	1.0	2.69	−0.379 ± 0.01	0.994	2.62
CCA3	−0.311 ± 0.003	0.999	2.69	−0.378 ± 0.02	0.994	2.62

**Table 3 materials-17-02514-t003:** Capacity of EDL and maximum specific capacity of PCM.

Capacity	PCM			
	CC	CCA1	CCA2	CCA3
*C_DEL_*, F/g	70.3	75.0	83.0	58.5
*C*, F/g	120	116	122	107
*C_DEL_*/*C*, %	59	65	68	55

**Table 4 materials-17-02514-t004:** EES parameters for the PCM/electrolyte/PCM electrochemical system.

Sample PCM	*L*, μH	*R*_s_, Ohm	CPE_1T_	CPE_1P_	*R*_1_, Ohm	CPE_2T_	CPE_2P_	*R*_2_, Ohm	*C*, F
CC	0.14	0.19	0.038	0.43	0.20	0.99	0.50	1.13	4.87
CCA1	0.19	0.39	0.004	0.75	0.07	1.05	0.53	0.85	4.90
CCA2	0.15	0.12	0.003	0.84	0.03	1.21	0.55	0.92	5.30
CCA3	0.13	0.20	0.007	0.71	0.05	0.80	0.56	1.70	5.10

## Data Availability

The original contributions presented in the study are included in the article, further inquiries can be directed to the corresponding authors.

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
