# Peer review of "Effect of Thermal Activation on the Structure and Electrochemical Properties of Carbon Material Obtained from Walnut Shells"

_materials, 2024, doi:10.3390/ma17112514_

Round 1
Reviewer 1 Report
Comments and Suggestions for Authors
Journal: Materials (ISSN 1996-1944)
Manuscript ID: materials-2950859
Type: Article
Title: Effect of Thermal Activation on the Structure and Electrochemical Properties of Carbon Material Obtained from Walnut Shells
Comments:
The present article discussed about the biomass derived carbon materials for the high-performance supercapacitor applications. The author has done some interesting work, but still it’s needs modify to publish in this journal. The manuscript could accept after the following minor revisions.
1. The abstract need to be better modified, some numerical values also to be added, it should present the core consequence of the work.
2. Introduction need to be better explained, the important and recent research in different activation temperature of carbon is to be included in the introduction. some old reference could be replaced with latest one. could be merged two or three paragraph.
3. The description of the used chemicals to be included clearly.
4. The surface area of the activated carbon is little lower than recently reported work, explain the reason.
5. The efficiency of recently reported materials for supercapacitor could be added as a new table.
The minor revision noted.
Comments on the Quality of English Languageminor revision needed.
Author Response
Please see the attachement

Reviewer 2 Report
Comments and Suggestions for Authors
Authors have obtained porous carbon material from walnut shells via thermal carbonization and subsequent thermal activation at 400 C for 1, 2, and 3 hours. And further investigated the electrochemical behavior. The work is well written and discussed keeping all aspects from synthesis, characterization, and electrochemical application. The reviewer is in favor of accepting the manuscript in MDPI Materials, following the minor revision.
In lines 83-84, the authors should provide some references, as well as should also mention the use of natural materials as a catalyst source for the carbon nanomaterials, (cite: https://doi.org/10.1016/j.diamond.2019.05.018; https://doi.org/10.1016/j.diamond.2021.108241).
To obtain the carbon materials, how was the process optimized for autoclave at 800 C and 400 C for thermal annealing?
What is the trend in the degree of graphitization and porosity if there is an increase or decrease in temperature?
How much was the yield of the product?
Lower the scale on the y axis in Figure 11, to have a clearer value of data points.
In certain parts, like the Raman section (lines 320-330), there exists a general discussion that may be omitted at the discretion of the authors.
Author Response
Dear Reviewer.
In the attachment you can find the answers to the review,
Best regards,
Authors

Reviewer 3 Report
Comments and Suggestions for Authors
I can see that there are already many articles published on walnut shell-derived carbon for supercapacitors. How does this work differ from those works? What new knowledge/observations does this work add to the literature?
Why did the authors choose 400 degrees as the optimum temperature? Did they conduct experiments to find the optimum? Generally, carbonization/activation is carried out at a temperature range of 600-900 degrees.
Why did the authors not go beyond the temperature range? Is there a specific reason reported, as they mention that increasing the duration reduces both thickness and lateral dimensions?
Even though the authors report that the majority of pores (75-78%) are mesopores, why did they only achieve low electrochemical performance?
The energy density and power density of the material should be calculated and reported.
Why are there so many reference citations at the end of the first paragraph in the introduction section?
What is the relevance of the following content where they mention composite electrodes, metals, and the electrolytic oxidation method? The connection to the background of this work is missing: "Several types of composite electrodes for batteries with improved performance characteristics were obtained by plasma electrolytic oxidation of valve group metals, particularly Ti [18,19]. In works [20,21,22], methods for ensuring the high quality of such composites and ways to optimize the process of their production were studied."
Multiple references can be cited for the following content: "Natural materials such as coconut shells, wood, resins, fossil coal, or synthetic materials such as polymers are typically used as precursors for AC production. The activation process develops the porous structure within the carbon material, creating micropores (< 2 nm), mesopores (2–50 nm), and macropores (> 50 nm) within the carbon grains."
No literature published in this field is discussed in the introduction.
"An analysis of the literature sources shows that researchers pay insufficient attention to the study of simple and affordable methods for obtaining and activating carbon material." Is this the reason why the authors undertook this work? Why did they study the activation duration? What is the purpose? Why did they study under limited air access? What could be the possible outcome? All of this needs to be discussed in the introduction.
"To obtain carbonized carbon material, the original bio-raw material (walnut shells) was introduced into an autoclave without additional grinding. The autoclave was then placed in a furnace and heated at a rate of 10 °C/min to a temperature of 800 °C, where it was held for 1 hour." What type of autoclave is it? Can it withstand 800 degrees? What is it made up of? Did the authors use any solvent or water in the autoclave?
"Thermal activation of this carbon was conducted at a temperature of 400 °C." This seems to be more like calcination than activation.
"To obtain carbonized carbon material, the original bio-raw material (walnut shells) was introduced into an autoclave without additional grinding." How much material was added? What is the yield of the final product? No information is given.
"The effectiveness of the process of grinding raw materials depends on understanding the mechanisms of fragmentation of shell structures, particularly by brittle fracture of the material near closable defects." This part should be expanded to include more information about the mechanism.
The flow of directing the content to the scope of this work, with the support of literature and background information, is completely missing in the introduction. Only the first paragraph of the introduction provides a proper base for the work. Other than the first paragraph, the rest of the introduction should be completely rewritten. A few recent research articles related to biomass-derived carbon materials for energy storage/supercapacitors should be referenced to observe and represent the flow of ideas and information.
In the XRD graph, what does it mean when it is mentioned as "(10)"?
Figures 2b and 2c have different aeromarks, maintain consistency.
Very little information is provided from the XRD. More information relating to and proving the graphitic nature of the material can be included.
Figures 3b, 3c, and 3d seem to correspond more to type 4 isotherm.
There is not much difference in the surface area with respect to the different duration of prepared materials obtained. The reason for this is not explained properly.
There is a sudden drop in the rate capability in the electrochemical studies. The reasoning should be explained in detail.
In Figure 11(a), the GCD curves at 10mA show a charging time of nearly 2500 seconds, whereas the discharge time is only nearly 1000 seconds. This will result in a very low coulombic efficiency, which is an important factor in determining the performance, life, and overall efficiency of the device. Repeat the experiments for the possible improvements and to find plausible underlying reasons. Explain the reasons in detail.
Why does only sample CCA2 show significantly low solution, charge transfer, and Warburg resistance compared to other samples? Explain the reasons in detail. Repeating the experiments might help.
Author Response
Dear Reviewer,
The answers to the review you can find in the attachment.
Authors

Round 2
Reviewer 3 Report
Comments and Suggestions for Authors
The manuscript can be accepted in its present form.